# DROPS: Deep Retrieval of Physiological Signals via Attribute-specific Clinical Prototypes

## Abstract

The ongoing digitization of health records within the healthcare industry results in large-scale datasets. Manually extracting clinically-useful insight from such datasets is non-trivial. However, doing so at scale while simultaneously leveraging patient-specific attributes such as sex and age can assist with clinical-trial enrollment, medical school educational endeavours, and the evaluation of the fairness of neural networks. To facilitate the reliable extraction of clinical information, we propose to learn embeddings, known as clinical prototypes (CPs), via supervised contrastive learning. We show that CPs can be efficiently used for large-scale retrieval and clustering of physiological signals based on multiple patient attributes. We also show that CPs capture attribute-specific semantic relationships.

## 1 Introduction

Physiological data are being collected at a burgeoning rate. Such growth is driven by the digitization of previous patient records, the presence of novel health monitoring and recording systems, and the recent recommendation to facilitate the exchange of health records (European Commission, 2019). This engenders large-scale datasets from which the manual extraction of clinically-useful insight is non-trivial. Such insight can include, but is not limited to, medical diagnoses, prognoses, or treatment.

In the presence of large-scale datasets, retrieving instances based on some user-defined criteria has been a longstanding goal within the machine learning community (Manning et al., 2008). This information retrieval (IR) process typically consists of a query that is used to search through a large database and retrieve matched instances. Within healthcare, the importance of an IR system is threefold (Hersh & Hickam, 1998; Hersh, 2008). First, it provides researchers with greater control over which patients to choose for clinical trial recruitment. Second, IR systems can serve as an educational and diagnostic tool, allowing physicians to identify seemingly similar patients who exhibit different clinical parameters and vice versa. Lastly, if the query were to consist of sensitive attributes such as sex, age, and race, then such a system would allow researchers to more reliably evaluate the individual and counterfactual fairness of a particular model (Verma & Rubin, 2018). To illustrate this point, let us assume the presence of a query instance that corresponds to a patient with an abnormality of the heart, atrial fibrillation, who is male and under the age of 25. To reliably determine the sensitivity of a model with respect to sex, one would observe its response when exposed to a counterfactual instance, namely the exact same instance *but* with a different sex label (Kusner et al., 2017). At present, deep-learning based IR systems within the healthcare domain fail to incorporate such patient-specific attributes.

Existing IR systems which retrieve instances from the electronic health records (Wang et al., 2019; Chamberlin et al., 2019) do not incorporate an attribute-specific search and do not trivially extend to physiological signals. In this paper, we propose to learn embeddings, referred to as clinical prototypes (CPs). CPs are efficient descriptors of a combination of patient-specific attributes, such as disease, sex, and age. We learn these embeddings via contrastive learning whereby representations of instances are encouraged to be similar to their corresponding clinical prototype and dissimilar to the others. To the best of our knowledge, we are the first to design a supervised contrastive learning based large-scale retrieval system for electrocardiogram (ECG) signals.

**Contributions.** Our contributions are the following:

- **Attribute-specific clinical prototypes** - we propose a supervised contrastive learning framework to learn embeddings, referred to as clinical prototypes (CPs), that are efficient descriptors of a set of patient attributes, e.g., disease, sex, and age.
- **Deep retrieval and clustering** - we exploit CPs to retrieve instances corresponding to a specific patient-attribute combination and assign instances to various clusters.

## 2 RELATED WORK

**Clinical representation learning** involves meaningfully representing clinical data for solving tasks. Most research attempts to learn representations of electronic health records (EHRs) (Miotto et al., 2016; Gee et al., 2019; Liu et al., 2019; Li et al., 2020b; Biswal et al., 2020; Darabi et al., 2020) in a generative manner. For example, Landi et al. (2020) and Huang et al. (2019) implement an autoencoder to learn patient representations. These representations are then clustered either in a hierarchical manner or via K-means. Other methods involve learning prototypes. For example, Li et al. (2020a) propose to do so via the ProtoNCE loss. Our approach, unlike theirs, exploits readily-available patient-attribute data and is not dependent upon the K-means algorithm. Moreover, Van Looveren & Klaise (2019) learn to perturb prototypes to derive interpretable counterfactual instances. Most similar to our work is that of Kiyasseh et al. (2020), which learns patient-specific representations while pre-training via contrastive learning, and Garnot & Landrieu (2020) where the distance between class prototypes learned in an end-to-end manner is regularized based on a pre-defined tree hierarchy. In contrast, we learn attribute-specific prototypes via supervised contrastive learning and capture their semantic relationships via distance-based regularization.

**Clinical information retrieval** whereby instances similar to a query are retrieved was first introduced in 1990 (Hersh & Greenes, 1990). Most research in this domain revolves around text (Gurulingappa et al., 2016; Wang et al., 2017; Rhine, 2017; Wallace et al., 2016). For example, D'Avolio et al. (2010) map text to SNOMED concepts to retrieve clinical documents. More recently, IR has been performed with biomedical images, and is referred to as content-based image retrieval (Saritha et al., 2019; Chittajallu et al., 2019). Others have extended this concept to EHR data (Goodwin & Harabagiu, 2018; Wang et al., 2019; Chamberlin et al., 2019). For example, Chamberlin et al. (2019) implement rudimentary IR methods such as divergence from randomness on the UPMC and MIMIC III (Johnson et al., 2016) datasets with the aim of discovering patient cohorts. In contrast to such methods, we implement a deep-learning based clinical information retrieval system for physiological signals.

## 3 METHODS

### 3.1 ATTRIBUTE-SPECIFIC CLINICAL PROTOTYPES

Information retrieval systems typically necessitate a query of some sort that is exploited to search through a large database and retrieve instances that satisfy criteria outlined by the initial query. Such a query can take on a multitude of forms (e.g., text, image, audio, etc.) depending on the modality of instances in the database. As we are primarily interested in large databases comprising physiological signals, we design a query that is based on such signals. Moreover, the type and specificity of instances that are retrieved highly depend on the criteria outlined by a query. In our context, these criteria comprise patient attribute information such as disease class, sex, and age. As a result, our query should be capable of retrieving physiological instances that are associated with the aforementioned patient attributes. To achieve this, we propose to learn a set of query embeddings, $P$, analogous to word embeddings in natural language processing, where $|P| = M$, representing each of the $M$ possible patient-attribute combinations within a dataset. Each embedding, $p_A \in P$, is an efficient descriptor of a set of attributes $A = \{\alpha_1, \alpha_2, \alpha_3\}$ where $\alpha_1 =$ disease class, $\alpha_2 =$ sex and $\alpha_3 =$ age. Given this attribute-specific interpretation, we refer to such embeddings as attribute-specific clinical prototypes or CPs. We propose to learn such CPs, in an end-to-end manner, via contrastive learning, as explained next.

### 3.2 LEARNING ATTRIBUTE-SPECIFIC CLINICAL PROTOTYPES

We assume the presence of a learner, $f_\theta : x \in \mathbb{R}^D \to v \in \mathbb{R}^E$ parameterized by $\theta$, that maps a $D$-dimensional input, $x$, to an $E$-dimensional representation, $v$. In information retrieval systems, a

reliable query should accurately retrieve instances that satisfy certain criteria. When dealing with representations, such reliability can be designed for by ensuring that the query embedding is in a similar subspace, and thus in proximity, to the representations of instances that satisfy said criteria. To achieve this proximity, we can attract query embeddings to representations in the database which share patient attribute information and repel them from those which do not. More specifically, we attract each representation of an instance associated with a set of attributes, $A$, to the clinical prototype, $p_A$, that describes that same set. This can be achieved in two ways.

**Hard assignment.** We encourage each representation, $v_i = f_\theta(x_i)$, of an instance, $x_i$, associated with a particular set of attributes, $A$, to be similar to the single clinical prototype, $p_A$, that describes the exact same set of attributes, and dissimilar to the remaining clinical prototypes, $p_j$, where $j \neq A$. We quantify this similarity, $s(v_i, p_A)$, using the cosine similarity, with a temperature parameter, $\tau_s$. By mapping each representation to a *single* CP, we refer to this as a hard assignment. To encourage this behaviour, one would optimize the following objective function for a mini-batch of size, $B$:

$$\mathcal{L}_{contra-hard} = -\frac{1}{B} \sum_{i=1}^{B} \log \left( \frac{e^{s(v_i, p_A)}}{\sum_j^M e^{s(v_i, p_j)}} \right) \quad (1) \qquad s(v_i, p_A) = \frac{v_i \cdot p_A}{\|v_i\|\|p_A\|} \cdot \frac{1}{\tau_s} \quad (2)$$

The hard assignment approach assumes a bijective relationship between each representation, $v_i$, and each clinical prototype, $p_A$. This many-to-one mapping implies that CPs without a perfect attribute match (i.e., a near miss) do not leverage potentially useful information from representations that exhibit some, albeit imperfect, overlap in patient attributes.

**Soft assignment.** To overcome the limitations of a hard assignment, we propose a soft assignment approach whereby each representation, $v_i$, is attracted to a *subset* of disease class-specific clinical prototypes, $L \subset P$. We opted for this class-specific setup to avoid erroneously attracting representations to CPs from a different class. This attraction would hinder retrievals based on disease class.

Recall, though, that the CPs in $L$ still describe different sets of non-class attributes. By attracting each representation to these CPs uniformly, CPs within a class will collapse to a single point, and thus be unable to distinguish between various patient attributes. We support this claim in Fig. 1a where we illustrate the t-SNE projection of the CPs for five different disease classes. To avoid such collapse, we modulate the attraction between each representation, $v_i$, associated with the attribute set, $A_i$, and each clinical prototype, $p_k$, associated with the attribute set, $A_k$. More specifically, we introduce a weight, $w_{ik}$, that is dependent on the discrepancy between the respective attribute sets, $d(A_i, A_k)$. This formulation, characterized by greater attraction between CPs and representations with more similar attribute sets, results in the CPs shown in Fig. 1b. Formally, we optimize the following objective function.

$$\mathcal{L}_{contra-soft} = -\frac{1}{B} \sum_{i=1}^{B} \left[ \sum_{k=1}^{M} \omega_{ik} \log \left( \frac{e^{s(v_i, p_k)}}{\sum_j^M e^{s(v_i, p_j)}} \right) \right] \quad (3)$$

$$\omega_{ik} = \begin{cases} \frac{e^{d(A_i, A_k)}}{\sum_j^{|L|} e^{d(A_i, A_j)}} & \text{if disease class}_i = \text{disease class}_k \\ 0 & \text{otherwise} \end{cases} \quad (4)$$

$$d(A_i, A_k) = [\delta(\text{disease class}_i = \text{disease class}_k) + \delta(\text{sex}_i = \text{sex}_k) + \delta(\text{age}_i = \text{age}_k)] \cdot \frac{1}{\tau_\omega} \quad (5)$$

where $\delta$ is the Kronecker delta function that evaluates to one if the argument is true and zero otherwise and $\tau_\omega$ is a temperature parameter that determines how soft the assignment is. For example, as $\tau_\omega \rightarrow 0$, the loss-term approaches the hard assignment formulation.

**Arrangement of clinical prototypes.** Clinical prototypes, learned in an end-to-end manner, as presented will exhibit a high and desirable degree of *inter-class* separability. However, prototypes *within* a class are still at risk of collapsing to a select few points. This would decrease their utility for attribute-based querying of instances. We would prefer to learn CPs that not only distinguish between different sets of attributes, but also reflect the semantic relationships between one another, as is done with word embeddings in natural language processing. Capturing semantic relationships is desirable as it allows for improved interpretability of the retrieval process. For example, representations that are equidistant from two CPs can indicate a graded difference in a particular attribute such as age. In Fig. 2 (left), we illustrate the desired arrangement of 16 CPs for two arbitrary classes.

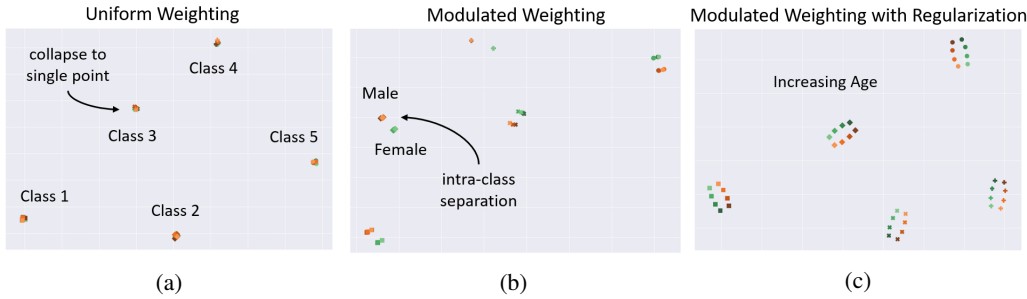

(a)           (b)           (c)

Figure 1: (a) Uniform weighting between each representation and CPs in the same class results in the collapse of CPs to a single point. (b) Proposed modulated weighting introduces some desirable CP intra-class separation. (c) Modulated weighting in addition to the regularization term exhibits healthy inter and intra-class separation in addition to reflecting the semantic relationships between CPs.

To arrive at this arrangement, we propose a distance-based regularization term, derived as follows. First, we use the $L_2$ norm to normalize all CPs in the set, $P$, and calculate the empirical pairwise Euclidean distance between them. As a result, we generate the matrix $\hat{D} \in \mathbb{R}^{M \times M}$ (Fig. 2 right). As we are only interested in the pairwise distances for prototypes *within* the same class, we focus on the sub-matrices, $\hat{D}_c \in \mathbb{R}^{M/C \times M/C}$ with $c \in [1, \ldots, C]$ where $C$ is the number of class-specific clusters. Each class' corresponding ground-truth sub-matrix, $D_c$, is populated with distances, $d_E$, that reflect the semantic relationships between CPs. We choose $d_E = \beta \times S$ where $\beta \in \mathbb{R}$ is a user-defined distance and $S \in \mathbb{Z}^+$ is the integer distance between attributes. For example, a pair of CPs that differ according to sex with $A_1 = \{\text{AFIB}, \text{M}, < 25\}$ and $A_2 = \{\text{AFIB}, \text{F}, < 25\}$, results in $S = 1$ and $d_E = \beta \times 1$. We then calculate the mean-squared error between $\hat{D}_c$ and $D_c$ $\forall c \in [1, \ldots, C]$. Our final objective function thus includes the soft contrastive loss (Eq. 3) and the regression loss.

$$\mathcal{L}_{reg} = \frac{C}{M^2} \sum_{c=1}^{C} (\hat{D}_c - D_c)^2 \qquad (6) \qquad \mathcal{L}_{comb} = \mathcal{L}_{contra-soft} + \mathcal{L}_{reg} \qquad (7)$$

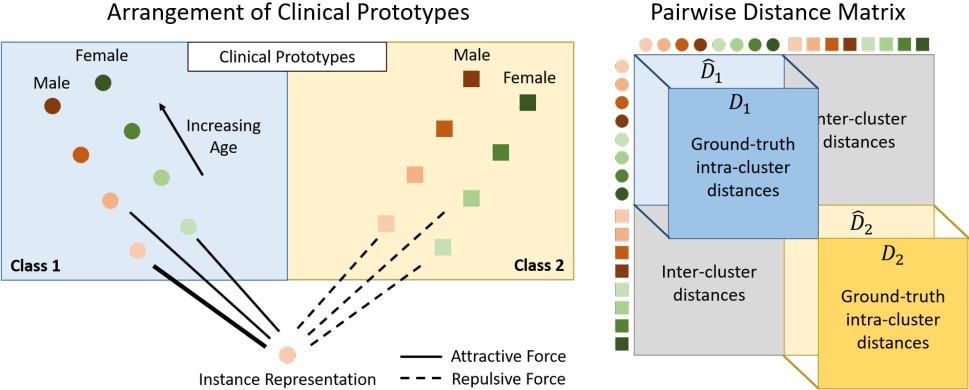

Figure 2: (**Left**) Visualization of the attractive and repulsive forces present between an instance representation and clinical prototypes that belong to two different clusters/classes. The clinical prototypes are coloured according to sex and shaded in ascending order of age. All repulsive forces are treated equally whereas attractive forces are weighted according to the degree of attribute matching between the instance and the respective clinical prototype. (**Right**) Matrix $\hat{D} \in \mathbb{R}^{|M| \times |M|}$ illustrating the Euclidean distance between each pair of clinical prototypes. The mean-squared error between the observed intra-cluster distances, $\hat{D}_c$, and the ground-truth intra-cluster distances, $D_c$, is minimized.

## 4 EXPERIMENTAL DESIGN

### 4.1 DATASETS

We conduct our experiments using PyTorch (Paszke et al., 2019) on two large-scale datasets that consist of physiological time-series, such as the electrocardiogram (ECG), alongside cardiac arrhythmia labels. **Chapman** (Zheng et al., 2020) consists of 12-lead ECG recordings from 10,646 patients alongside labels which we group into 4 major classes. **PTB-XL** (Wagner et al., 2020) consists of 12-lead ECG recordings from 18,885 patients alongside labels which we group into 5 major classes (Strodthoff et al., 2020). Further details can be found in Appendix A.

### 4.2 DEPLOYING CLINICAL PROTOTYPES FOR RETRIEVAL

We aim to exploit clinical prototypes to retrieve physiological instances, that satisfy certain criteria, from a large database. One could argue that a simple text-based query, instead of CPs, in modern databases will allow for the retrieval of appropriate instances. However, this only holds if all instances in the database are labelled. Although we have used, as an exemplar, readily-available attributes such as sex and age, our method extends to other attributes which cannot be trivially searched in a database. In other words, the maximal utility of retrieval via CPs is realized when the troves of instances in the database are *unlabelled*. To reiterate, the clinical motivation for such retrieval is multi-fold. First, it allows researchers to discover patients that may be suitable for clinical trial recruitment. This, in turn, may allow pharmaceutical companies to trial their drugs on a more diverse patient cohort that is more representative of the population. Second, CPs can be deployed across datasets from different clinical institutions, for example, to retrieve patient sub-cohorts that satisfy a set of attributes. These patient sub-cohorts can now be further analysed to identify sub-cohort variability, which in turn can guide future clinical treatment. With that in mind, to reliably and quantitatively evaluate the retrieval capabilities of the CPs, we must have access to the ground-truth attributes associated with the instances in the held-out set.

To perform retrieval, we treat the CPs as the query set and retrieve the closest instances ($\downarrow$ Euclidean distance) from a held-out dataset. In evaluating this task of retrieval, we deploy a commonly used metric known as Precision at $K$ (P@$K$). This metric quantifies whether at least one of the $K$ retrieved instances matches the query. In our context, however, a match can occur according to any of the three attributes: class, sex, and age. Therefore, we extract a single attribute, $\alpha_m$, from the set of attributes, $A$, for each of the $M$ clinical prototypes, and the same attribute, $\alpha_i$, for each of the $K$ retrieved instances, and define the following *attribute-specific* P@$K$.

$$\text{P@}K(\alpha) = \frac{1}{M} \sum_{m=1}^{M} \delta \left( \sum_{i=1}^{K} \delta(\alpha_i = \alpha_m) \geq 1 \right) \qquad (8)$$

### 4.3 DEPLOYING CLINICAL PROTOTYPES FOR CLUSTERING

We also exploit clinical prototypes to cluster instances. In this setting, and in contrast to the retrieval setting, we treat each representation of an instance as a query and search across the CPs to assign each representation a set of attributes. From this perspective, CPs can be thought of as multiple centroids of a cluster which are used to label representations. To evaluate these clusters, we first calculate the pairwise Euclidean distance between each representation and CP before assigning each representation the specific attribute, $\alpha_i^{pred}$, of the CP to which it is closest. Given the ground-truth attribute for each representation, $\alpha_i^{true}$, we can calculate the accuracy, Acc($\alpha$), of the assignments. We can also quantify the agreement between the attribute assignments of the ground-truth, $\vec{\alpha}^{true} = \{\alpha_{true}^1, \alpha_{true}^2, \ldots, \alpha_{true}^N\}$, and those obtained via clustering, $\vec{\alpha}^{pred} = \{\alpha_{pred}^1, \alpha_{pred}^2, \ldots, \alpha_{pred}^N\}$, by calculating the adjusted mutual information AMI($\alpha$) $\in [0, 1]$.

$$\text{Acc}(\alpha) = \frac{1}{N} \sum_{i=1}^{N} \delta(\alpha_i^{pred} = \alpha_i^{true}) \qquad (9)$$

$$\text{AMI}(\alpha) = \frac{\left[ \text{MI}(\vec{\alpha}^{true}, \vec{\alpha}^{pred}) - \mathbb{E}(\text{MI}(\vec{\alpha}^{true}, \vec{\alpha}^{pred})) \right]}{\mathbb{E}(\text{H}(\vec{\alpha}^{true}), \text{H}(\vec{\alpha}^{pred})) - \mathbb{E}(\text{MI}(\vec{\alpha}^{true}, \vec{\alpha}^{pred}))} \qquad (10)$$

where $\mathrm{MI}(\vec{\alpha}^{true}, \vec{\alpha}^{pred})$ represents the mutual information between the ground-truth and the predicted set of attributes, and $\mathrm{H}(\vec{\alpha})$ represents the entropy of the set of attributes.

## 4.4 BASELINES

Where appropriate, we compare our method to the following approaches:

- **K-Means** implements Expectation Maximization to arrive at class-specific centroids. Instances are then assigned to the class of their nearest centroid. We perform K-Means a) on the input instances (**K-Means Raw**) and b) on representations of instances learned using Eq. 7 (**K-Means Combined**).

- **Mean Representations** takes the mean of the representations, learned via Eq. 7, that belong to a particular attribute combination and treats such mean representations as the centroids of the clusters.

- **DeepCluster (DC)** (Caron et al., 2018) is an iterative method that performs K-Means on representations, pseudo-labels instances according to their assigned cluster, and then exploits such labels for supervised training. The final set of instance labels is taken from the epoch with the lowest validation loss.

- **Information Invariant Clustering (IIC)** (Ji et al., 2019) maximizes the mutual information between the posterior class probabilities assigned to an instance and its perturbed counterpart. We adapt IIC to the time-series domain by perturbing instances with additive Gaussian noise, $\epsilon \sim \mathcal{N}(0, \sigma)$. We also incorporate auxiliary over-clustering as it was shown to significantly improve performance. The final set of instance labels is chosen by taking the argmax of the output probabilities.

- **SeLA** (Asano et al., 2020) pseudo-labels instances by implementing the Sinkhorn-Knopp algorithm to allow for supervised training. We pseudo-label instances after each epoch of training and use prior class information to determine the number of clusters, which should boost performance.

- **Deep Transfer Cluster (DTC)** (Han et al., 2019) calculates the distance between representations and cluster prototypes to generate a probability distribution over classes. The KL divergence between this distribution and a target distribution is minimized.

## 4.5 HYPERPARAMETERS

We chose the temperature parameters, $\tau = 0.1$, as per (Kiyasseh et al., 2020), and $\tau_w = 1$. The regression regularization term (Eq. 6) requires the specification of $\beta$. We chose $\beta = 0.2$ due to our choice of distance metric (squared Euclidean distance) and the number of attribute groups. If $\beta$ were too small in magnitude, the intra-cluster separability of clinical prototypes would be too small. If $\beta$ were too large, then the clinical prototypes from *different* classes would begin to overlap with one another and class separability diminishes. For both datasets, sex $\in \{M, F\}$, and age is converted to quartiles. For the Chapman and PTB-XL datasets, $|\text{class}| = 4$ and 5, respectively, and thus $M = |\text{class}| \times |\text{sex}| \times |\text{age}| = 32$ and 40, respectively. We use a network with 1D convolutional modules to generate representations. Further network and implementation details can be found in Appendix B.

# 5 EXPERIMENTAL RESULTS

## 5.1 DEPLOYING CLINICAL PROTOTYPES FOR RETRIEVAL

We exploit CPs to retrieve attribute-specific physiological signals from a database. In this section, we task the CPs with retrieving the closest $K = [1, 5, 10]$ instances in a held-out set and, in Table 1, illustrate the resulting P@$K$. We also stratify the results according to the number of matched attributes between the query (CPs) and retrieved instances. For example, # of matched attributes = 3 implies that a perfect match has occurred for all attributes: class, sex, and age.

We find that the Mean Representation approach is most likely to retrieve physiological instances that satisfy the desired patient attribute criteria. For example, on Chapman, Mean Representation

Table 1: Retrieval performance on Chapman and PTB-XL datasets for $K = 1, 5, 10$. Results are stratified according to the number of matched attributes between the query (CP) and retrieved instances and are averaged across 5 seeds. Bold indicates best performing method for retrieving instances with perfect attribute match.

| Method | # of matched attributes | Chapman | | | PTB-XL | | |
|---|---|---|---|---|---|---|---|
| | | $K = 1$ | 5 | 10 | 1 | 5 | 10 |
| DTC (Han et al., 2019) | $\geq 1$ | 71.9 | 100 | 100 | 70 | 90 | 100 |
| | $\geq 2$ | 25 | 71.9 | 90 | 22.5 | 52.5 | 80.5 |
| | $= 3$ | 3.1 | 15 | 36.9 | 2.5 | 9.5 | 16.5 |
| Mean Representation | $\geq 1$ | 91.9 | 97.5 | 100 | 99.0 | 100 | 100 |
| | $\geq 2$ | 55 | 79.4 | 90 | 71.5 | 94.5 | 100 |
| | $= 3$ | **10.6** | **23.8** | **36.9** | **15.5** | **32** | **43.5** |
| DROPS Combined | $\geq 1$ | 76.3 | 99.4 | 100 | 70 | 100 | 100 |
| | $\geq 2$ | 26.9 | 74.4 | 93.8 | 19.5 | 60 | 82 |
| | $= 3$ | 2.5 | 12.5 | 21.3 | 2.5 | 7 | 10 |

achieves a $P@(1) = 91.9$ whereas DTC and DROPS Combined achieve $P@(1) = 71.9$ and 76.3, respectively, when evaluated based on at least one attribute match. This implies that 91.9% of the representations used as the query set were able to retrieve a physiological instance with at least one attribute match. In retrieval settings, however, we are typically interested in perfect attribute matches (i.e., # of matched attributes = 3). In our context, the superior performance of Mean Representation extends as the # of matched attributes = $1 \rightarrow 3$. For example, on Chapman, Mean Representation achieves a $P@(1) = 10.6$ whereas DTC and DROPS Combined achieve $P@(1) = 3.1$ and 2.5, respectively. Recall that with Mean Representation, we are exploiting the average of the representations, learned via our method, that belong to a particular attribute combination as the query set. These mean representations can be visualized in Fig. 4 (top row). With this mind, the aforementioned findings illustrate the potential utility of our supervised contrastive learning setup in obtaining rich representations for attribute-specific retrieval. We hypothesize that the poorer performance of DROPS Combined relative to Mean Representation is due to the more extreme embedding characterized by the former's clinical prototypes. This can be seen in Fig. 4 (top row). Although such clinical prototypes, which are farther from the class decision boundary, may be beneficial for retrieving instances from the same class, they will perform worse on other attributes.

To qualitatively evaluate clinical prototypes' retrieval capabilities, we choose CPs at random to form the query set, calculate their Euclidean distance to the representations of instances in a *held-out* dataset, and retrieve the 5 which are closest. In Fig. 3, we visualize the ECG signals that correspond to these closest representations and colour their borders green if their class label matches that of the query, and red otherwise. We can see that at least 60% of the retrieved instances match the class label.

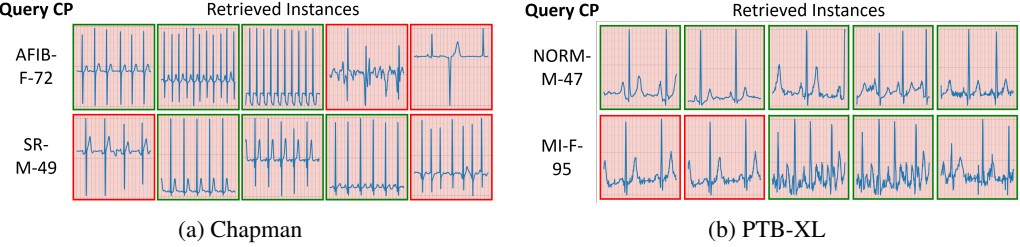

(a) Chapman          (b) PTB-XL

Figure 3: Top 5 retrieved instances based on query CPs from the (a) Chapman and (b) PTB-XL datasets. Green and red borders indicate retrieved instances whose ground-truth class attribute matches that of the CPs and those whose ground-truth class attribute does not match, respectively.

## 5.2 DEPLOYING CLINICAL PROTOTYPES FOR CLUSTERING

So far, we have illustrated the utility of exploiting clinical prototypes for retrieval purposes. In this section, we evaluate the utility of CPs for clustering. In Table. 2, we illustrate the clustering performance of DROPS relative to that of state-of-the-art clustering methods.

There are several takeaways from Table 2. First, we find that our supervised contrastive learning paradigm generates rich representations. This can be seen by the superiority of K-Means Combined relative to K-Means Raw where the $\text{Acc}(\text{class}) = 81.7$ and $29.2$, respectively. Note that although the former approach learns clinical prototypes, it does *not* yet exploit them directly for clustering. This, in turn, allows for the reliable evaluation of the representations alone. We provide further evidence in support of the richness of these representations in Sec. 5.3. When we directly leverage clinical prototypes for clustering, as is done with DROPS, we find that this further improves clustering performance. For example, DROPS Combined achieves an $\text{Acc}(\text{class}) = 90.3$ and $76.0$ on the Chapman and PTB-XL datasets, respectively, which outperforms not only its K-Means counterpart but also recent state-of-the-art methods such as DTCluster, DeepCluster, and SeLA.

To further illustrate the utility of clinical prototypes, we compare DROPS Combined to the Mean Representation associated with each of the attribute combinations. We find that that the former ↑ $\text{Acc}(\text{class}) \approx 10\%$ on both Chapman and PTB-XL. We hypothesize that this discrepancy is partially due to the asymmetric definition of the contrastive objective function (Eq. 3) in which negative samples are only considered from the set of clinical prototypes and not from the representations. Nonetheless, the aforementioned findings underscore the vital role clinical prototypes play in clustering. We also note that our method can be trivially incorporated into DeepCluster, for example, to generate pseudo-labels. Lastly, in Table 2b, we show that DROPS is flexible enough to *simultaneously* cluster according to *multiple* attributes, a feature that does not trivially extend to other methods. Such flexibility can provide researchers with improved control over sensitive attributes.

Table 2: Clustering performance of representations in the validation set based on (a) class attribute and (b) sex and age attributes. Results are shown across 5 seeds.

(a) Cardiac arrhythmia class attribute

| Method | Chapman | | PTB-XL | |
|---|---|---|---|---|
| | Acc | AMI | Acc | AMI |
| *Unsupervised Clustering* | | | | |
| K-Means Raw | 29.2 | 0.3 | - | - |
| SeLA (Asano et al., 2020) | 21.0 | 10.2 | 10.5 | 1.6 |
| IIC (Ji et al., 2019) | 27.0 | 0.2 | 22.0 | 0.5 |
| DC (Caron et al., 2018) | 32.3 | 3.0 | 10.5 | 4.9 |
| *Supervised Clustering* | | | | |
| DTC (Han et al., 2019) | 79.8 | 58.8 | 67.3 | 22.1 |
| K-Means Combined | 81.7 | 61.9 | 36.5 | 27.4 |
| Mean Representation | 80.3 | 65.0 | 64.7 | 29.1 |
| DROPS Contrastive | 89.8 | 72.1 | 76.1 | 36.0 |
| DROPS Combined | **90.3** | **72.8** | **76.0** | **35.9** |

(b) Sex and age attributes

| Method | Chapman | | PTB-XL | |
|---|---|---|---|---|
| | Sex | Age | Sex | Age |
| DTC (Han et al., 2019) | 54.8 | 25.3 | 58.6 | 25.7 |
| Mean Representation | 54.8 | 30.3 | 69.7 | **39.4** |
| DROPS Contrastive | 56.8 | 37.4 | **74.3** | 19.3 |
| DROPS Combined | **57.4** | **38.0** | 73.5 | 19.5 |

## 5.3 CAPTURING SEMANTIC RELATIONSHIPS BETWEEN CLINICAL PROTOTYPES

Up until now, we have shown that clinical prototypes can be successfully deployed for retrieval and clustering. We encouraged such prototypes to exhibit inter and intra-cluster separability, with the aim of capturing semantic relationships between them. In this section, we look to confirm whether such semantic relationships are indeed captured. In Fig. 4 (top row), we illustrate the t-SNE projection of representations in the training set alongside the *average* class-specific CP. In Fig. 4 (bottom row), we illustrate the t-SNE projection of the CPs colour-coded and shaded according to sex and age groups, respectively. For clarity, we follow the same coding scheme presented in Fig. 2 (left).

We show that our training paradigm leads to clinical prototypes that can satisfy the semantic relationships between the class, sex, and age attributes. This can be seen by the high separability between, and ordering of, the sex and age attributes in Fig. 4 (bottom row). This observation holds regardless

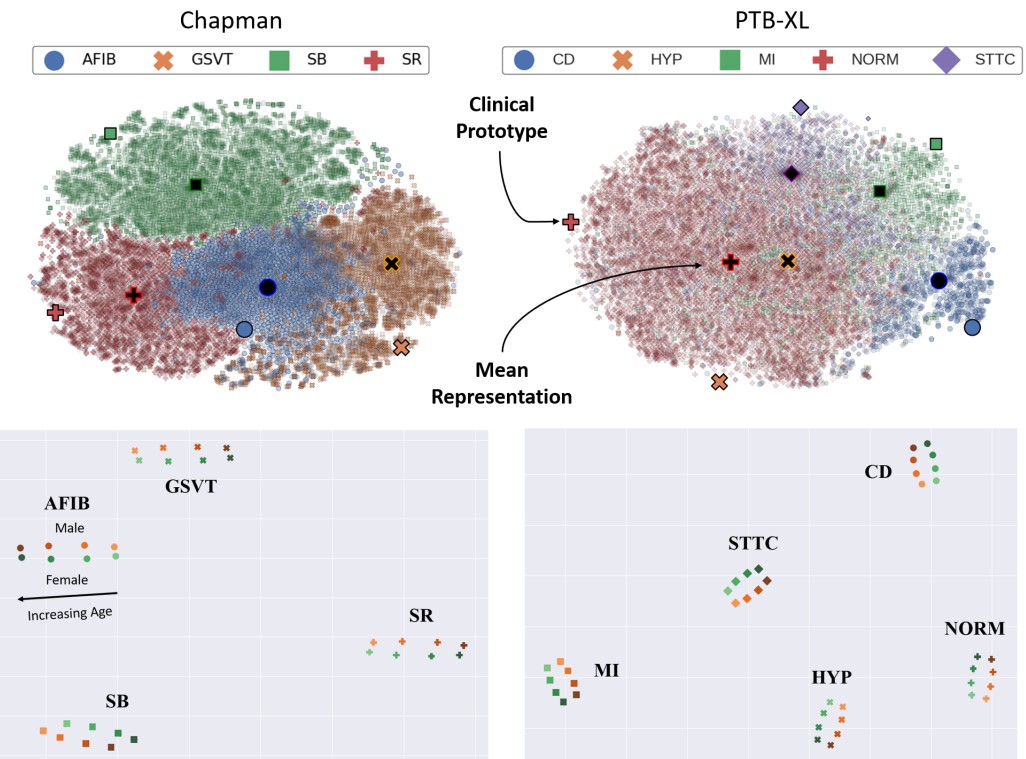

Figure 4: **(Top Row)** t-SNE projection of the representations in Chapman and PTB-XL, respectively. These are shown alongside the average clinical prototype and the mean representation *per class* (shown in larger size). **(Bottom Row)** t-SNE projection of the corresponding clinical prototypes colour-coded and shaded according to sex and age, respectively (see Fig. 2 left).

of the class attribute or the dataset that is experimented with. We further note the high degree of similarity between these projections, empirically-derived, and expected (see Fig. 2 left). We claim that the adoption of this formation by the CPs is driven primarily by our weighting mechanism (Eq. 3) and distance-based regularization term (Eq. 6).

## 6 DISCUSSION AND FUTURE WORK

In this paper, we propose a supervised contrastive learning framework, DROPS, for the retrieval and clustering of physiological signals. In the process, we learn representations, entitled clinical prototypes, that efficiently describe the state of a patient associated with a set of attributes, e.g., disease, sex, and age. We show that representations learned via DROPS can be exploited to retrieve instances that reflect desired criteria. Moreover, we show that that clinical prototypes, in addition to capturing semantic relationships between patient attributes, can simultaneously cluster physiological instances based on multiple attributes. We now elucidate several avenues worth exploring.

**Learning disentangled clinical prototypes.** CPs are attribute-specific representations. Disentangling these representations into their constituent attributes might facilitate the interpretability of CPs and their application to fair ML. Preliminary exploratory work is performed in Appendix C.

**Incorporate continuous attributes.** CPs are currently limited to discrete attributes. Scaling CPs to a continuous set and large number of attributes will allow for more fine-grained retrieval and further increase their utility.

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
