# OpenReview forum: "DROPS: Deep Retrieval of Physiological Signals via Attribute-specific Clinical Prototypes"
_ICLR.cc/2021/Conference — Reject_

### Official Review · AnonReviewer3 · 2020-10-31
**Review of the submission called DROPS: Deep Retrieval of Physiological Signals via Attribute-specific Clinical Prototypes**

**Rating:** 2
**Confidence:** 4

**Review:**

The manuscript focuses on clinical natural language processing of electronic health records. More precisely, it addresses a text classification task called information extraction or named entity recognition from these clinical records. Its contributions include developing an embedding model to capture clinical prototypes (CPs), via supervised contrastive learning, and presenting experimental evidence of these learnt CPs capturing attribute-specific semantic relationships and being helpful in subsequent clinical natural language processing task of information retrieval and clustering of clustering of physiological signals. I find this text processing methodology interesting, carefully described, and supplementary to other studies.

However, unfortunately, the authors demonstrate limited understanding of the related literature. First, the second and third paragraph of the Introduction section have many sentences that require references to be inserted. Second, and more importantly, the Related work section seems to not capture the key papers and trends of the field (I suggest reading some systematic reviews or surveys on clinical natural language processing, information extraction, and information extraction by, for example, Wendy Chapman, Carol Friedman, and Pierre Zweigenbaum), and, for example, as illustrated by the ImageCLEF and CLEFeHealth evaluation labs, computer vision tends to proceed faster than text analytics (see, e.g., https://www.researchprotocols.org/2018/7/e10961/), and not the other way around as claimed by this section.

In addition, to feel convinced of the presented experimental evidence, I would have wanted to see statistical significance tests, confidence intervals, effect sizes, or similar presented. I could not find this methodology described in the manuscript or its outcomes, although the narrative repeatedly referred to significant performance gains. Please clarify.

As my main minor comment, I would like to see a clearer separation of the materials, methods, and experiments sections from results, as well as including clearer justifications of this study design. For example, the aforementioned significance topic has not been addressed sufficiently. Another illustration of somewhat difficult task for the reader is to understand the experimental design as a whole and be convinced of this study being rigorous is the Experimental Results section including a lot of methodological details as opposed to only obtained outcomes.

I also suggest including a conclusion statement as well as embedding more evidence (e.g., evaluation materials and methods plus obtained indicators of performance gains, such as measure values and effect sizes) to convince the reader in the abstract. To continue, please remember to punctuate equations and formulae.

Finally, typically the Methods, Experiments, and Experimental Results sections would have been written using a past tense to emphasise a finished (as opposed to an ongoing) study where materials, methods, and experiments have already been chosen, justified, and completed. Most importantly, please avoid having inconsistent tense in these sections (see, e.g., Section 4.5. using a past tense whereas almost all others are having a present tense).

---

> ### Author Response · Authors · 2020-11-20
> **Response to Reviewer 3 - Round 1**
>
> **THIS IS NOT AN NLP PAPER**
> We would like to urge the reviewer to please re-read the manuscript in its entirety. This is because we believe the reviewer has either 1) not read the manuscript in the first place or 2) had given it a cursory glance. We arrive at this conclusion based on their feedback which claims that we have designed an natural language processing (NLP) method focused on named entity recognition and clinical information extraction. Moreover, our method does not experiment with electronic health records as the reviewer had suggested. We exclusively work with physiological signals and more specifically, the electrocardiogram (ECG).
>
> **HIGH-LEVEL OVERVIEW**
> In the event the reviewer does not read the entire manuscript, we provide a high-level overview of what our paper entails:
> 1) We are focused on the retrieval and clustering of physiological signals (e.g., the electrocardiogram) in large databases.
> 2) We learn clinical prototypes (embeddings, and NOT word embeddings) that summarize a set of patient attributes such as disease class, sex, and age, via supervised contrastive learning.
> 3) We leverage, and illustrate the utility of, clinical prototypes to retrieve physiological instances from two large-scale ECG datasets.
> 4) We leverage, and illustrate the utility of, clinical prototypes to cluster physiological instances based on a multitude of patient attributes.
> 5) We show that our clinical prototypes capture the semantic relationships between their corresponding patient attributes.
> 6) The benefit of such retrieval and clustering is that is allows for the discovery of patients that may be suitable for clinical trials, and thus increasing streamlining clinical trial workflows. It can also allow for the discovery of similar patients both within and across datasets from different healthcare institutions, and thus open the door for patient sub-cohort discovery. This, in turn, can guide the way physicians treat their patients.
>
> We hope the reviewer takes the time and effort to read the manuscript and provides us with valuable feedback that can be used to augment the original manuscript.

---

### Official Review · AnonReviewer5 · 2020-11-05
**This work proposes to use a contrastive learning method to learn attribute specific prototypes of clinical physiological signals and show their utility in the tasks of information retrieval and clustering  while also showing these learned representations capture attribute specific semantic relationships .**

**Rating:** 4
**Confidence:** 4

**Review:**

##########################################################################
Reasons for score:

The authors propose a novel contribution of designing a contrastive learning based attribute specific retrieval system for physiological signals, but the experiments in the paper aren’t sufficient in showing the proposal’s utility relative to appropriate baselines for the two tasks specified ( IR and clustering ) and  don’t discuss the prior work the physiological signal representation mechanism is based on which is central to the work.

The work could be greatly improved by clarifying some important aspects of the proposed system and by conducting more appropriate experiments to back the result claims made.


##########################################################################
Pros:
1) The authors propose a novel contribution of designing a contrastive learning based attribute specific retrieval system for physiological signals  and do a fairly good job of showing how it differs in regards to existing works ( with the exception of 1 or 2 important ones )
2) The contrastive learning approach of the contrastive-soft loss is motivated and explained well and shown to improve
upon solely using K-Means on representations learned via the L_contrastive−soft + L_regression  combined loss objective.

##########################################################################
Cons:

1) There are no comparisons for how the system performs in the IR case relative to other works and the baselines for the clustering case are not appropriate ( as they are all unsupervised ) and this method explicitly needs supervision via attributes to derive attribute specific prototypes for learning.

   Also, in the retrieval section the use of R@K doesn’t seem appropriate as used.
   Usually R@K means does your result (document / entity / etc ) appear in the top K results, but here its different.

   For instance, if we look at the sex attribute, we have a set of M total attributes one per prototype (either 32 or 40) that is split
   evenly between male and female [ M, F, M , F … ]  =  [a1, a2, a3, a4 .. aM ] for all M combinations of attributes.
   Given my understanding of the formulation in 4.2, we take a particular CP (lets say its the 15th of 32 ), then find its K nearest neighbors by euclidean distance, and then compare the true sex of those K returned neighbors with that of the CP.
   However this “sex” attribute is either M or F and doesn’t pertain to the exact sex prototype of CP 15 ( ie, the task isn’t find the correct 1 out of 32, but rather find any of the correct 16 out of 32 ).. Is this correct?

   In the medical field, the need for exact matches seems like it would preclude just expanding the number of returned items K in hopes of finding one correct results especially when there is a simple possibility for querying with structured data anyways ( ie filter signals by age / sex / class first )  and then just use retrieval of similar signals that way.  Is that incorrect?

    In section 5.1 60% at K=5 is hard to assess for quality without comparing it against other retrieval methods.

    Specific to the clustering section, the external baselines described are all unsupervised and not directly comparable to the
    supervised DROPS or K-means Combined methods as such.  Effectively the work shows that doing k-means on representations learned by the paper is not as good in terms of ACC/AMI as compared with those learned from two proposed proposed variants of contrastive learning.

  The two proposed variants of attribute specific contrastive learning are interesting, but hard to size up against other methods.
    For instance  taking latent space reps from the  autoencoder + prototypical classifier network  from Gee et al19 and comparing how K-Means on those representations does with K-means Combined or DROPS would be directly comparable and appropriate ( there are probably other supervised learning methods to construct representations that you could use from the prior works section as well ).


2) Kiyasseh et al., 2020 “ Clocs: Contrastive learning of cardiac signals”  is not mentioned in the prior work section and mentioned once in section 4.5 to discuss the use of a hyper parameter.
	This work seems quite similar in the methods, datasets, etc used particularly in regards to contrastive learning of cardiac signals for representation learning so in the background section the authors should clarify and say if this is an extension to that work or explain the differences.

3) Points that were unclear to me:
	3.1 The construction of D_hat and the ground truth D in section 3.3 in “Arrangement of clinical prototypes” .  It seems that in order to regularize with respect to the CPs, in order to calculate pairwise distances between them to make Dhat, you need to be given the prototypes initially ( otherwise where does the ground truth D come from ).  Its very possible this is a misunderstanding on my end, but this section could have been clearer.

       3.2 Similarly, is C set to 4 and 5 for the two datasets specifically or are they set to 8 in both cases ( sex |2| x age |4| = |8| )?  The wording left that a little unclear to me and specifying that in section 4.5 would resolve that.

	3.3 How are the "v" representations and prototypes actually learned?   The architecture given in section C of the supplement outputs an E x C, which seems its intended for the prototypes, but its not entirely clear.

       3.4  There are two different architectures used for the Chapman and PLB-XL datasets and the reason given is the size of the later,
            but the first has ~10k examples ( each of size 2500 ) and 4 classes whereas the later has 18k each with size 2500 and 5 classes so they seem to be relatively similar in size no?

4)  The work Gee et al 19 “Explaining Deep Classification of Time-Series Data with Learned Prototypes”  ( ICML 19 https://arxiv.org/abs/1904.08935 ) seems that it should possibly be cited in the prior works section since it learns prototypes on clinical signals and shows how the t-sne representations of the prototypes lie with respect to the learned latent representations of the patient clinical signals.  The work doesn't explicitly describe a retrieval mechanism, but it does assign classes to the prototypes by finding the nearest neighbor in euclidean distance over the training set and giving the prototype its class which is similar in nature to the retrieval mechanism proposed here ( though with a different and non-contrastive learning mechanism ).

5)  If the Beta parameter is distance metric dependent then it would be useful to show an experiment showing what occurs in terms of ACC / AMI by adjusting this parameter to justify setting it to .2.

6) Table 1 this table should really be  split to be attribute specific  ( have additional columns for age / sex / class for each K=1 , K=5 and K=10 for the two datasets)
    It seems like the approach would do best at the class task and then do worse ( as is 5.1 ) at getting the sex or age attributes, but without that information we can’t discern this for the retrieval task ( though its shown for the clustering task in Table 2 )

7) In Table 2 , the Combined method doesn’t improve performance in (a) or (b) for the PTB-XL case and it would be useful to hear discussion on that.

8) Below Table 2, I disagree with the statement “Note that this approach does not yet exploit clinical prototypes” since the training setup which learns these representations in fact does leverage CPs for learning purposes  since they are explicitly used in the contrastive soft loss objective.

9) K-Means could be used to provide clustering by different attributes ( though possibly not simultaneously ) and adding that to Table 2 (b) could help elucidate if its advantageous or not to learn the clusters simultaneously in fact.

10) In Section 5.3, the contrastive learning representations are separated pretty well in the Chapman set, but it seems to a much lesser extent in the PTB-XL data.  Why is that?
   What does plotting the k-means centroids of these representations with T-SNE give?
   How would this differ from setting K=32 ( for the Chapman case ) and then using the average of the representations fo each
   example in a specific attribute combination grouping ( sex - age - class ) as the centroid for each K?

##########################################################################

Questions during rebuttal period:
Please address and clarify the cons above

#########################################################################

---

> ### Author Response · Authors · 2020-11-20
> **Response to Reviewer 5 - Round 1**
>
> We thank the reviewer for taking the time and effort to review the manuscript and for providing us with valuable feedback. We address your comments below.
>
> **RETRIEVAL BASELINES**
> DROPS learns clinical prototypes that can be exploited for the retrieval and clustering of physiological instances associated with multiple attributes. To enable a fair comparison of DROPS to other methods, we would need to experiment with those that are capable of performing retrieval based on *multiple* attributes. **Would it be possible for the reviewer to recommend several baseline methods that are believed to be appropriate for our scenario?**
>
> **CLUSTERING BASELINES**
> The original implementations of the baselines chosen for the clustering experiment (Sec. 5.2) are indeed completely unsupervised. However, when we adapted such methods for our experiments, we did include weak-supervision. Such weak supervision varied from one method to the next. For example, when implementing DeepCluster, we chose the value of K (the number of clusters) to optimize for in the K-Means implementation, based on the ground-truth number of disease classes that we had in the dataset. Granted, this supervision is weaker than that provided by a supervised cross-entropy loss, for example, yet is still existent.
>
> Having said that, we are currently in the process of conducting experiments with a supervised variant of a deep clustering algorithm (https://openaccess.thecvf.com/content_ICCV_2019/papers/Han_Learning_to_Discover_Novel_Visual_Categories_via_Deep_Transfer_Clustering_ICCV_2019_paper.pdf). We will be including these results in Table 2 of Sec. 5.2.
>
> **RETRIEVAL EVALUATION METRIC**
> We thank the reviewer for mentioning the R@K metric. To clarify, we believe our description of the retrieval process, its implementation, and the results are appropriate. However, the metric that we describe and are actually reporting is the Precision @ K, and not the Recall @ K, as we had included the original manuscript. To see why, take the following example; let us say the attribute of interest is sex and the sex attribute associated with CP # 15 is male. We retrieve the K nearest neighbours to CP # 15 that are in the validation set. We then ask the question ‘Does the sex attribute of at least one of these K nearest neighbours match the sex attribute of CP # 15?’. We then repeat the process for all M clinical prototypes. The reason this is considered Precision @ K is that we are quantifying how many of the retrieved instances happen to have a male sex attribute. In contrast, with Recall @ K, we would be quantifying what percentage of all instances with a male sex attribute were actually retrieved. We have reflected these changes in the modified version of the manuscript.
>
> **SIMPLE QUERY OF DATABASE**
> We have responded to a similar question posed by Reviewer 6. We include that response here for convenience. If the held-out set is already labelled with attributes such as disease class, sex, and age, then a clinician/researcher with access to a database can quite simply search for a physiological signal that matches specific attributes. In such a scenario, embeddings would not be needed. This is not the scenario that we are interested in. Instead, we are interested in the more realistic scenario characterized by a clinical institution which has access to a large, yet unlabelled, database of physiological signals. Although we have used relatively straightforward attributes that may be easily accessible, our method trivially extends to attributes which are more difficult to obtain. Nonetheless, to retrieve physiological signals that satisfy specific attributes from such a database, the clinician/researcher can now leverage the learned CPs to do so. Importantly, in order for us to reliably and quantitatively evaluate the retrieval capabilities of the CPs, we must have access to the ground-truth attributes associated with the instances in the held-out set. As for the clinical motivation for such retrieval, it is multi-fold. First, it allows researchers to discover patients that may be suitable for clinical trial recruitment. This, in turn, may allow pharmaceutical companies to trial their drugs on a more diverse patient cohort that is more representative of the population. Second, CPs can be deployed across datasets from different clinical institutions, for example, to retrieve patient sub-cohorts that satisfy a set of attributes. These patient sub-cohorts can now be further analysed to identify sub-cohort variability, which in turn can guide future clinical treatment. Please refer to Sec. 4.2 in the modified version of the manuscript to see the appropriate changes.
>
> Response is continued in Part 2.

---

> > ### Author Response · Authors · 2020-11-20
> > **Response to Reviewer 5 - Round 1 (Part 2)**
> >
> > **GEE ET. AL PAPER**
> > Our retrieval experiment is different to what is done in Gee at al. https://arxiv.org/abs/1904.08935. In their paper, classes are assigned to the prototypes based on nearest neighbours. We are performing the exact opposite. We are assigning classes to the retrieved instances based on the attributes associated with the prototype. They also have an exclusive classification head for making predictions and do not explore the utility of their prototypes in the context of retrieval nor clustering, which are the emphasis of our paper. Lastly, the number of prototypes they learn is also based on some arbitrary value, m. In our case, it is based on the number of attribute combinations of interest in the dataset and is thus more physiologically grounded/gives you more control over the instances we are retrieving.
> >
> >  **CLARIFICATION OF DISTANCE MATRIX**
> > We have rewritten Sec. 3 in order to improve clarity. We hope our modified description of the clinical prototypes can clarify the construction of the distance matrix. In a nutshell, for each class, we have two matrices: 1) an empirical pairwise distance matrix ($\hat{D}$) based on the actual distances between clinical prototypes and 2) a ground-truth distance matrix (D) which we populate with whatever distance we like. We choose to populate it with distances that reflect the semantic relationships between clinical prototypes. By minimizing the mean-squared error between $\hat{D}$ and D, then the *actual* clinical prototypes will begin to exhibit the desired semantic relationships.
> >
> > **CLARIFICATION OF NUMBER OF PROTOTYPES**
> > We have clarified the number of prototypes that we are learning for each dataset in Sec. 4.5.
> >
> > **CLARIFICATION OF ARCHITECTURE**
> > We thank the reviewer for pointing this out. Our original description of the architecture erroneously included a classification head, which mapped from representations to classes. To clarify, we do not need this head in our implementation as we are performing clustering based solely on the representations. We have since corrected this architecture which can be found in the modified version of the supplementary material.
> >
> > **TSNE OF PTBXL**
> > The relatively poorer separability of the t-SNE projections of representations in the PTB-XL dataset is not necessarily reflective of the limitations of our method. Rather, it is indicative of a more difficult task to solve relative to the one introduced by the Chapman dataset. This is likely compounded by the fact that if we were to use a more heavily parameterized network, then we might obtain improved separability. Regardless, the purpose of Fig. 4 (top row) is to illustrate the locations of the learned clinical prototypes relative to the representations of instances.
> >
> > **EXPLOITATION OF CLINICAL PROTOTYPES**
> > We would like to clarify our statement “Note that this approach does not yet exploit clinical prototypes” found below Table 2. We agree that clinical prototypes were used in the training procedure and thus influence the resultant representations. However, if we are being more precise, what we mean is that clinical prototypes in the K-Means Combined approach were not directly used for clustering purposes. In contrast, with DROPS, we are explicitly using the clinical prototypes as 'centroids' in order to cluster the representations.
> >
> > **CLINICAL PROTOTYPES VS. AVERAGE REPRESENTATIONS**
> > When plotting the t-SNE of the average representation from each attribute combination, we find that the location of these average representations differ from that of the clinical prototypes which we showed in Fig. 4. This could be due to the asymmetric nature of the loss shown in Eq. 3.
> >
> > **ARCHITECTURE**
> > We have primarily chosen a more heavily parameterized network for the PTB-XL dataset relative to the Chapman dataset due to both the relatively larger size of the PTB-XL dataset and also the relatively higher difficulty of the cardiac arrhythmia classification task of the latter. We found that our chosen architectures led to reasonable performance for our datasets, and do not dispute the fact that additional architecture tuning is bound to result in improvements.
> >
> > We hope the above responses and the modified version of the manuscript have addressed your concerns.

---

### Official Review · AnonReviewer6 · 2020-11-06
**Missing many details and flawed retrieval experiment design**

**Rating:** 4
**Confidence:** 3

**Review:**

##########################
Summary:

This paper proposes to tackle the problem of retrieving and clustering physiological signals by learning clinical prototypes via supervised contrastive learning. Three readily available patient attributes, disease, age, and sex, are used to assist the learning. A hard assignment of samples to prototype is proposed, and further relaxed to a soft one to utilize the samples that do not have exactly matched attribute set. A regularization term is also proposed to encourage intra-cluster distances. Two ECG datasets are used to evaluate the proposed model.

##########################
Pros:
The target problem to tackle is important and the idea of using prototype learning approach is interesting.


##########################
Cons/Questions:
1. Overall, the writing can be improved. I find several places confusing and many details are not fully described. For example, in Section 3.1, "To that end, we propose to learn a set of embeddings, $p\in P$", what is space $P$?
2. It's unclear to me what needs to be learned, $p$ or $v$, or both?
3. In Eq (4), what is $\text{class}_i$? is it already known?
4. In Eq (8), the $\delta(\cdot)$ function is undefined. Is it the indicator function?
5. For the retrieval task, why the CPs are used as the query set to retrieve instances from the held-out set, instead of the reverse (given unseen sample as query and retrieve samples from database)?
6. For the retrieval task, it is unclear to me if the attributes are available for the held-out set. If yes, then the evaluation metric of the retrieval task seems weird to me: If the task is to simply retrieve samples with similar attributes, it is straightforward to just query the attributes and just retrieve the samples with a maximal number of matched attributes. Why do we need to learn embeddings for such task? If the attributes are not available, what is the clinical motivation for retrieving such instances?
7. Usually the ECG signals are processed with CNN or RNN models, it's unclear if and what models are used by the authors to generate the feature $x$ for the input ECG.

---

> ### Author Response · Authors · 2020-11-20
> **Response to Reviewer 6 - Round 1 (Part 1)**
>
> We thank the reviewer for taking the time and effort to review the manuscript and for providing us with valuable feedback. We address your comments below.
>
> **CLARIFYING PROTOTYPES**
> We have modified the methods section (Sec. 3) to improve clarity. Please refer to Sec. 3 of the modified version of the manuscript to see these changes. In a nutshell, clinical prototypes can be thought of as being analogous to 'word embeddings' in the natural language processing (NLP) literature. CPs are parameters that are randomly initialized and updated via gradient descent as is typically done with 'word embeddings'. In our setup, we are learning two sets of parameters. The first set of parameters consists of the network parameters, theta. These are used to parameterize the function, f, which maps instances to representations, v. These representations are thus not directly learned. The second set of parameters consists of the clinical prototypes (CPs), which we denote as p. By optimizing the objective function in Eq. 7 through gradient descent, we jointly learn the network parameters, theta, and the clinical prototypes, p.
>
> **CLARIFYING TERMINOLOGY**
> Recall that our proposed method is a supervised contrastive learning method. As an exemplar, in our paper, these supervisory signals arise from patient-specific attributes such as disease class, sex, and age. Therefore, in Eq. 4, class_i is in reference to the disease class (more specifically, the cardiac arrhythmia) associated with the particular physiological signal. We clarify this equivalence between disease and class in the modified version of the manuscript. Generally, though, any arbitrary patient-specific attribute could have been used as a replacement for the class attribute (e.g., in the event cardiac arrhythmia labels are unavailable).
>
> **KRONECKER DELTA FUNCTION**
> Yes δ(⋅) is the indicator function, which we refer to as the Kronecker delta function. We had defined this function in Sec. 3.3 (paragraph 5) right after using it for the first time in Eq. 5.
>
> **JUSTIFICATION FOR RETRIEVAL TASK**
> We realize that CPs can be exploited in various ways. The first way we exploit CPs is by treating them as the query set to retrieve instances from the validation set. The motivation for doing so stems from the potential application of CPs in a clinical setting. Take the following example; a clinical institution may have access to a large physiological database. From this physiological database, clinicians/researchers have a desire to intelligently retrieve specific instances (examples) that satisfy particular attributes. To enable such retrieval, they have a query set of attribute-specific clinical prototypes that they can deploy. However, one can reliably depend on such CPs to retrieve appropriate instances only once they have been validated on an external dataset. This validation is exactly encompassed by the experiment we describe in Sec. 4.2 and whose results are shown in Sec. 5.1. We have modified the manuscript to justify our use of CPs as a query set.
>
> **LABELLED HELD-OUT SET**
> As you mentioned, if the held-out set is already labelled with attributes such as disease class, sex, and age, then a clinician/researcher with access to a database can quite simply search for a physiological signal that matches specific attributes. In such a scenario, embeddings would not be needed. This is not the scenario that we are interested in. Instead, we are interested in the more realistic scenario characterized by a clinical institution which has access to a large, yet unlabelled, database of physiological signals. Although we have used relatively straightforward attributes that may be easily accessible, our method trivially extends to attributes which are more difficult to obtain. Nonetheless, to retrieve physiological signals that satisfy specific attributes from such a database, the clinician/researcher can now leverage the learned CPs to do so. Importantly, in order for us to reliably and quantitatively evaluate the retrieval capabilities of the CPs, we must have access to the ground-truth attributes associated with the instances in the held-out set. As for the clinical motivation for such retrieval, it is multi-fold. First, it allows researchers to discover patients that may be suitable for clinical trial recruitment. This, in turn, may allow pharmaceutical companies to trial their drugs on a more diverse patient cohort that is more representative of the population. Second, CPs can be deployed across datasets from different clinical institutions, for example, to retrieve patient sub-cohorts that satisfy a set of attributes. These patient sub-cohorts can now be further analysed to identify sub-cohort variability, which in turn can guide future clinical treatment. Please refer to Sec. 4.2 in the modified version of the manuscript to see the appropriate changes.
>
> Response continued in Part 2.

---

> > ### Author Response · Authors · 2020-11-20
> > **Response to Reviewer 6 - Round 1 (Part 2)**
> >
> > **ARCHITECTURE**
> > In Appendix C, we had outlined the neural network architecture used to extract the representations, v. We use a three layer CNN for the Chapman dataset in order to remain consistent with more recent papers that leverage contrastive learning in the context of physiological signals. For the PTB-XL dataset, we leverage a modified ResNet-18 where the number of blocks per layer is reduced from two to one. We chose a more highly parameterized network for the PTB-XL dataset given its increased size and the relative difficulty of the task.
> >
> > We hope the above responses and the modified version of the manuscript have addressed your concerns.

---

### Decision · Program_Chairs · 2021-01-07
**Final Decision**

**Decision:**

Reject

**Comment:**

This paper proposes to learn clinical prototypes via supervised contrastive learning to facilitate the reliable retrieval of clinical information and clustering in large datasets. The presentation of the paper could be substantially improved – e.g., the overview and motivation of the paper, the definition of clinical prototypes, selection of certain evaluation criteria, clarification of terminology in equations, the description of the motivations and settings of the experiments, etc.  In addition to the need to substantial improvement in clarity, major concerns include lack of comparison with more supervised approaches and discussion of relevant literatures raised by reviewers.